# Vitamin D Deficit as Inducer of Adenotonsillar Hypertrophy in Children with Obstructive Sleep Apnea—A Prospective Case-Control Study

**DOI:** 10.3390/children10020274

**Published:** 2023-01-31

**Authors:** Pietro De Luca, Arianna Di Stadio, Pasquale Marra, Francesca Atturo, Alfonso Scarpa, Claudia Cassandro, Ignazio La Mantia, Antonio Della Volpe, Luca de Campora, Domenico Tassone, Angelo Camaioni, Ettore Cassandro

**Affiliations:** 1Otolaryngology Department, San Giovanni-Addolorata Hospital, 00184 Rome, Italy; 2Otolaryngology Department, AORN Moscati, 83100 Avellino, Italy; 3Otolaryngology Department, University of Catania, 95124 Catania, Italy; 4Department of Medicine, Surgery and Dentistry, University of Salerno, 84084 Salerno, Italy; 5Department of Surgical Sciences, University of Turin, 10124 Turin, Italy; 6Cochlear Implant and Middle Ear Unit, Santobono-Posilipon Hospital, 80122 Naples, Italy

**Keywords:** vitamin D3, hypovitaminosis D, vitamin D deficiency, adenotonsillar hypertrophy, obstructive sleep apnea

## Abstract

(1) Objective: This prospective case-control study aimed to assess the level of serum vitamin D comparing pediatric non-allergic patients with obstructive sleep apnea (OSA) and healthy controls. (2) Methods: The period of the enrollment was from November 2021 to February 2022. Children with uncomplicated OSA caused by adenotonsillar hypertrophy (ATH) were recruited. Allergy was excluded by skin prick test (SPT), and the determination of serum IgE level using ELISA test. Plasma concentration of 25-hydroxy vitamin D (25-OHD) was quantitatively determined; then, the vitamin D concentration in patients was compared with healthy controls matched for sex, age, ethnicity, and characteristics. (3) Results: Plasma 25-OHD levels were significantly lower in patients than in healthy subjects (mean 17 ng/mL, 6.27 DS, range 6–30.7 ng/mL, vs. mean 22 ng/mL, 9.45 DS, range 7–41.2 ng/ ml; *p* < 0.0005). The prevalence of children with vitamin D deficiency was significantly higher in the ATH group than controls. The plasma 25-OHD level did not change following the ATH clinical presentation (III or IV grade according to the Brodsky scale), while the different categories of 25-OHD status (insufficiency, deficiency, and adequacy) in the ATH group were statistically significantly different (*p* < 0.001) from healthy controls. (4) Conclusions: This study identified statistically significant differences between the ATH group and control regarding the plasma concentration of vitamin D; this data, despite not being directly linkable to the lymphoid tissue hypertrophy (*p*-value not significant), might suggest a negative effect of vitamin D deficit on the immune system.

## 1. Introduction

Palatine tonsils and adenoids, located in the upper respiratory and gastrointestinal tract, are part the Waldeyer’s lymphatic ring; these structures are fundamental for inducing a local immune response and producing antibodies. Commonly, in the pediatric population, these structures are hypertrophic. Adenotonsillar hypertrophy (ATH) is one of the most common pathological conditions in the pediatric population causing chronic nasal obstruction, recurrent otitis media and recurrent sinusitis, and rhinorrhea [1,2]; moreover, ATH determines an obstruction of the upper respiratory tract (URT) with consequent snoring and obstructive sleep apnea syndrome (OSA). The latter can negatively impact patients’ quality of life [3].

Although the pathogenetic mechanisms that induce ATH are unclear, some studies suggested that suffering from an allergy exposes the patient to a high risk of developing ATH [4]. On the other hand, several studies hypothesized that low serum levels of vitamin D could be implicated in the pathogenesis of chronic infections of the Waldeyer’s ring [5,6]; this vitamin has both immunostimulant and immunomodulating properties [7], so its deficit might expose people to recurrent viral infections of the URT and be responsible for a massive uncontrolled immune answer in the Waldeyer ring [8].The recurrent infection and the massive production of cells in the lymphoid tissue could explain the increased volume of adenoids and tonsils. Hypovitaminosis D affects children and adolescents worldwide, and recent evidence identified an association of this deficit with several metabolic disturbances [9]. Vitamin D seems to have an important role both on the systemic and local immune answer. In addition, most children with dental problems, such as periodontal disease, present a low level of vitamin D; some reports link a vitamin D deficit to a decrease/loss of the alveolar process bone and a decrease/loss of connective tissue attachment [10]. These oral disorders, such as sleep bruxism, temporomandibular joints disorders, and dental caries, common among the pediatric population, impact the quality of sleep [11].

Recently, Shin et al. [6] found a correlation between low levels of 25-OHD and ATH in children; the authors performed multiple allergen simultaneous tests (MAST) to evaluate the atopic status, and more than half of all children with SDB had a vitamin D deficit. The major limit of this study was its retrospective nature that made it difficult to clarify the real cause–effect link between ATH and vitamin D level. Moreover, despite the authors identifying a negative correlation (Spearman test) between adenoid obstruction, tonsil hypertrophy, and BMI, and vitamin D levels, they did not perform a multivariate analysis to study the effect of all these variables. Finally, the gender differences were not considered.

By excluding allergy as responsible for ATH (this is the first case-control study where clinicians excluded allergy both by skin prick test (SPT), and by serum IgE level determination using ELISA test), we speculate that a vitamin D deficit might have a role in persistent chronic ATH. To validate this hypothesis, we conducted a prospective study measuring the level of vitamin D in the serum of a cohort of non-allergic pediatric patients with ATH, and healthy control, and comparing these results. A correlation test and a multivariate analysis were performed to understand the impact of multiple factors on ATH.

## 2. Materials and Methods

A prospective case-control study was conducted at the “San Giovanni di Dio e Ruggi D’Aragona” Hospital in Salerno, Italy, from November 2021 to February 2022. Children (age range 4–14 years) suffering from uncomplicated OSA due to ATH (patients group) were recruited in the Otolaryngology Department. The study protocol was approved by Institutional Review Board of “San Giovanni di Dio e Ruggi D’Aragona Hospital”, and informed written consent was obtained from the patients’ parents. Controls were healthy children not affected by ATH, and matched for age, sex, ethnicity, and month of blood testing.

Inclusion criteria: age 0–16 yr, with results of home sleep apnea test (sleep study with ApneaLink Air (ResMed, Germany) which consisted of a nasal cannula detecting flow and a pulse oximeter for revealing O2 saturation during sleep and heart rate). Parameters assessed were Apnea Hypopnea Index (AHI) corresponding to average of apneas and hypopneas per hour of sleep; Oxygen Desaturation Index (ODI) which was expression of the number of desaturation events (>3%) per hour of sleep; and O2 NADIR (the lower O2 saturation level) [12,13], supportive of OSA diagnosis (apnea–hypopnea index greater than 1, or a minimum oxygen saturation of less than 92 percent) [14] (results of home sleep apnea test are summarized in Table 1).

Exclusion criteria: known inherited or acquired immune diseases, celiac disease, renal diseases, special diet (vegan, milk-free diet), chronic diseases (type 1 diabetes and inflammatory bowel disease), genetic disorders (Down syndrome, allergy), malnutrition (BMI < 18), the use of vitamin D supplements, neoplasms, and severe mental disorders.

We provide a completed Strengthening the Reporting of Observational Studies in Epidemiology (STROBE) checklist for the study [15].


**
*Collection and sample processing*
**


Participants were non-fasting. Venous samples were collected from the antecubital vein (±6 mL) into a Vacutainer tube^®^ (Becton Dickinson). The samples were allowed to clot for 30 min, and centrifuged for 10 min at 3000× *g*/min at 4 °C. The venous sera were stored as 500 μL aliquots. Serum samples were stored at −80 °C until analysis, a condition in which 25-OHD is stable.


**
*Clinical assessment*
**


The study flow-chart is explained in Figure 1.

At admission, clinical data including age, sex, anthropometric data (body weight and BMI), number of upper respiratory infections (URIs) per year, and comorbidities were collected. Physical examination was performed by two different ENT specialists with fiberoptic flexible nasopharyngoscopy; the size of tonsils was evaluated using the Brodsky scheme (from grade 0 to grade 4; in this scale, 0 indicates that the tonsils do not impinge on the airway, 1+ indicates less than 25% airway obstruction, 2+ indicates 25–50% airway obstruction, 3+ indicates 50–75% airway obstruction, and 4+ indicates more than 75% obstruction); by endoscopy, the volume of retronasal lymphoid tissue (adenoids) and the one on the base of the tongue was evaluated [16].

Presence of allergy was excluded by SPT, and measurement of blood IgE level by ELISA.

Plasma concentration of 25-hydroxy vitamin D (25-OHD) was quantitatively determined by chemiluminescent-immunoassay (CLIA) (LIAISON, DiaSorin, Saluggia, Italy). Vitamin D status was defined according to the ESPGHAN criteria as follows: severe deficiency (<10 ng/mL), deficiency (<20 ng/mL), and sufficiency/adequacy (≥20 ng/mL) [17]. The same procedure was performed for the control group.


**
*Outcome measures*
**


Primary outcome measures were the plasma 25-hydroxy vitamin D (25-OHD) levels in patients and controls, and prevalence of children with vitamin D deficiency (<20 ng/mL) in both groups. Secondary outcome measures were (i) correlation between plasma 25-OHD levels and age in both groups, (ii) the prevalence of children suffering from vitamin D deficit according to BMI and to ATH clinical presentation, and (iii) the effect of gender, age, BMI, and vitamin D levels on ATH.


**
*Statistical analysis*
**


Results are reported as means ± standard deviation (SD). MedCalc Statistical Software version 14.8.1 (MedCalc Software Ltd., Ostend, Belgium; http://www.medcalc.org; 2014) was used to perform statistical analysis. Within-group and between-group analysis were performed. Pearson (P) and Spearman (S) tests were used to identify the correlation between BMI and vitamin D level, age and vitamin D level, and ATH score and the level of vitamin D in the ATH group. Point-biserial was used for sex and vitamin D. Same tests, except ATH, were performed in the control group. Multilinear regression analysis was performed to evaluate the impact of sex, age, and vitamin D level on ATH scores. τ test was used to compare the numerical values between the ATH group and healthy controls. Chi-square (χ) was performed to compare the nominal variable (deficit, insufficiency, and adequate) between the groups. *p*-value < 0.05 was considered for statistical significance.

## 3. Results

### 3.1. Within Group

#### 3.1.1. Patients Group

66 ATH participants (mean age 6.7 yr, range 4–14 yr, SD 2.219) were included following the inclusion and exclusion criteria. Among these, 20 were females (30.3%), with a mean age of 7.2 yrs (range 4–14 yrs), and 46 males (69.7%), with a mean age of 6.4 yrs (range 5–11 yrs). Of the prospective participants, 20 patients were excluded following the exclusion criteria (five suffered from celiac disease, three suffered from renal disease, three had a special diet, three had an associated chronic conditions (type 1 diabetes and inflammatory bowel disease), two were affected by genetic disorders (Down syndrome), and four had an allergy).

Of the participants, 14 children (21.2%) were diagnosed of IV grade of ATH, 52 children (78.8%) with III grade of ATH. Only three children (4.7%) had hypertrophy of the adenoid tissue in the base of the tongue. Common symptoms were snoring (*n* = 43, 65.1%), nasal obstruction, (*n* = 50, 75.7%), mouth breathing (*n* = 58, 87.9%), and poor school performance (*n* = 10, 15.1%). In the ATH group, 212 patients were affected by URIs.

We did not identify a statistically significant correlation between BMI and vitamin D level (p = 0.9; r = 0.01233), sex and vitamin D (p = 0.3), or age and vitamin D level (p = 0.1; r = −0.1895). No statistically significant correlation was identified between vitamin D level and ATH scores (p = 0.1; r= 0.01233).

Multilinear regression analysis identified, despite the absence of statistically significant p, only that the level of vitamin D was slightly correlated with ATH scores (r = 0.19).

#### 3.1.2. Control Group

The control group included 64 healthy children (mean age 6.9 years, range 4–14, SD 2.45): 25 females (39%), with a mean age of 7.0 years (range 4–13), and 39 males (61%), with a mean age of 6.9 years (range 4–14). In the control group, 55 children (85.9%) were diagnosed with I grade of ATH; nine children (7.8%) with II grade of ATH. Only one child (4.7%) was affected by hypertrophy of the lymphoid tissue of the base of the tongue. Of these, 89 children in the control group were referred to as being affected by URI.

We did not identify a statistically significant correlation between BMI and vitamin D level (p = 0.8; r = 0.02331), sex and vitamin D (p = 0.2), or age and vitamin D level (p = 0.1; r = 0.1895). Demographic characteristics are reported in Table 2.

**Table 2 children-10-00274-t002:** Characteristic of patients with ATH and healthy controls.

Variable	ATH (*n* = 66)	Healthy Controls (*n* = 64)	*p*
**Mean age ± SD (range), yr**	6.7 ± 2.21 (4–14)	6.9 ± 2.45 (4–14)	NS
**Gender—no (%)**			
*Female*	20 (30.3%)	25 (39%)	NS
*Male*	46 (69.7%)	39 (61%)	NS
**Mean BMI**	17 (SD 4.6)	17.5 (SD 3.4)	NS
**Clinical presentation (palatine tonsil)—no. (%)**			/
*I grade*		55 (85.9%)	
*II grade*		9 (7.8%)	
*III grade*	52 (78.8%)		
*IV grade*	14 (21.2%)		
**Clinical presentation (adenoid tissue)—no. (%)**			/
*I grade*		55 (85.9%)	
*II grade*		9 (7.8%)	
*III grade*	52 (78.8%)		
*IV grade*	14 (21.2%)		
**Clinical presentation (base of the tongue hypertrophy)—no. (%)**			/
*Yes*	3 (4.7%)	1 (1.6%)	
*No*	63 (95.3%)	63 (98.4%)	
**Upper respiratory infections/yr—no.**	212	89	/
**Mean 25-OHD ± SD (range)—ng/mL**	17 ± 6.27 (6–30)	22 ± 9.45 (7–41)	<0.0005
**25-OHD status—no. (%)**			
Insufficiency	8 (12.1%)	4 (6.2%)	<0.001 ^a^
Deficiency	38 (57.6%)	20 (31.2%)	<0.001 ^a^
Adequacy	20 (30.3%)	40 (62.6%)	<0.001 ^a^
**25-OHD deficiency according to BMI class—no. (%)**			
Underweight	4 (8.7%)	2 (8.3%)	NS ^b^
Normal weight	39 (84.8%)	22 (91.7%)	NS ^b^
Overweight	3 (6.5%)	0	/
Obese	0	0	
**25-OHD deficiency according to clinical presentation**			
III grade	17.47 ± 5.82 (6.3–30.7)	/	/
IV grade	15.24 ± 6.25 (6–25.1) [*p* = 0.2147 (NS)]		

**ATH—**adenotonsillar hypertrophy; **SD—**standard deviation; **BMI—**body mass index; **25-OHD—**25-hydroxy vitamin D; **NS—**not significant. ^a^ comparison of 25-OHD levels between 25-OHD status classes in both groups; ^b^ comparison of 25-OHD levels between BMI classes in both groups.

### 3.2. Between Groups

#### 3.2.1. Anthropometric Data

No statistically significant differences were found between children with ATH and controls in anthropometric measurements (*p* = 0.1). The mean BMI was 17 (SD 4.6) in children with ATH and 17.5 in the control group (SD 3.4). The prevalence of weight categories was similar in both group (Table 1).

#### 3.2.2. Vitamin D Status

Table 1 shows 25-OHD results in ATH children and controls. Plasma 25-OHD levels were significantly lower in patients than in healthy subjects (mean 17 ng/mL, 6.27 DS, range 6–30.7 ng/mL, vs. mean 22 ng/mL, 9.45 DS, range 7–41.2 ng/mL; *p* < 0.0005). The percentage of children with vitamin D deficiency was significantly higher in the ATH group than controls (*n* = 40, 60.6% vs. *n* = 29, 45.3%). The plasma 25-OHD level did not change according to ATH clinical presentation (III or IV grade according to the Brodsky scale), while the different categories of 25-OHD status (insufficiency, deficiency, and adequacy) in the ATH group were statistically significantly different (*p* < 0.001) from those of the healthy control.

## 4. Discussion

Overall, our results showed that patients with ATH had a vitamin D level statistically significantly lower than the control group. Although we did not identify statistically significant correlations between the level of vitamin D and ATH (small sample, low variability between the ATH score in the sample), we speculate that the reduced level of vitamin D might negatively impact the immunological answer. This might explain the difference in the clinical features observed between patients and healthy controls. To the best of our knowledge, this is the first prospective case-control study that explored the correlation between hypovitaminosis D and ATH performed on children, in whom allergy was excluded by the skin prick test plus serum IgE level.

ATH is a common disease in children, and the most common cause of chronic breathing obstruction and risk factor for developing OSA in pediatric population. Tonsillectomy was first described 3000 years ago in the Hindu literature, and was realized by the Roman doctor Cornelio Celsus in the first century before Christ, using his bare hands.

Previously, the main indication for tonsillectomy was chronic or recurrent tonsillitis; today, this surgery is performed for the treatment of sleep-disordered breathing (SDB) and obstructive sleep apnea. Many children are treated by surgical adenotonsillectomy, despite some authors proposing partial intracapsular/extracapsular tonsillectomy with several devices (microdebrider, monopolar or bipolar forceps electrocautery, plasma ablation) as a safe and effective technique for treating ATH and OSA; the supports of these technologies affirm that they ensure less complications in terms of hemorrhage, post-operative pain, and infections compared to traditional adenotonsillectomy, with a very low 5-year tonsillar regrowth [1]. Despite these promising results, partial tonsillectomy is not yet recommended by the guidelines developed by the American Academy of Otolaryngology—Head and Neck Surgery Foundation [18].

Although the cause of ATH has not been completely understood, allergy seems to be involved in the hypertrophic process. Cho et al. reported that children with ATH have been identified as sensible to several allergens that impact the serum and/or adenotonsillar tissue immune answer; the children had a higher rate of specific immunoglobulin E in the AT tissue than in serum; furthermore, the prevalence of asthma and allergic rhinitis were significantly higher in children with local atopy than control [19]. To exclude this confounding factor, we screened all the children (both ATH groups, both healthy controls) for allergy (serum IgE level and prick tests), and only children who had a negative result to both tests were included in the study.

Vitamin D is a fat-soluble vitamin, mainly synthesized in the body through ultraviolet B (UVB) exposure on the skin or taken orally through food and/or supplements. According to the definition of the Endocrine Society, we can define the following categories: deficiency (<20 ng/mL); insufficiency (between 20 and 29 ng/mL); and sufficiency (≥30 ng/mL). Vitamin D deficiency/insufficiency is a global epidemic, estimated to affect over one billion people worldwide, including children.

Recently, Salepci et al. [20] identified an association between OSA and vitamin D deficiency in adults, and Kheitandish-Gozal et al. [21] and Ozgurhan et al. [22] showed a relationship between low levels of vitamin D and the risk of suffering from OSA. The role of vitamin D in the ATH (which represents the most important risk factor for OSA in children) is still unclear and there is no consensus on the association between hypovitaminosis D and pediatric ATH. Prono and colleagues [23] investigated the association between vitamin D levels and sleep disorders, including OSA, supposing that low levels of the vitamin could play a role in the pathogenesis of these disorders; although the exact mechanism by which the vitamin D level impacts sleep regulation is still unclear, the presence of vitamin D receptors (VDRs) in areas of the brainstem, which are involved in sleep regulation, could explain the reason why a decrease of this vitamin in the blood induces sleep alterations. These VDRs are widely expressed in (i) the prefrontal cortex, which is activated during Non-Rapid Eye Movement (NREM) and deactivated during Rapid Eye Movement (REM) sleep, and (ii) the cingulate gyrus, which lies immediately above the corpus callosum, and its continuation in the cingulate sulcus; the latter is activated by breathing variations and blood pressure changes which are common in sleep apnea.

Di Stadio et al. [8], by a randomized controlled trial on patients affected by ATH, using a supplement containing 400 mg of vitamin D, showed the reduction of tonsillar volume and episodes of tonsillar infection in those patients who were assuming the supplement compared to control. Despite the authors not evaluating the vitamin D levels, the use of the supplement improved the clinical findings. The authors speculated that the improvement of clinical findings was determined by the immune-stimulating and immunomodulating effect of the supplement. These data might indirectly support our hypothesis of evaluating the effect of a low vitamin level on the volume (expressed as score) of the adenotonsillar hypertrophy.

In fact, we speculate that the hypertrophy of adenoidal and tonsillar lymphatic tissues might be a consequence of the low 25-OHD serum levels, because low levels of this vitamin downregulate the answer of the immune system. The reduction of the systemic immune answer exposes these children to an increased risk of suffering from viral infection. The viruses stimulate the local immune answer in the Waldeyer ring causing the hypertrophy of the lymphoid tissue.

According to this hypothesis, we should expect lower 25-OHD levels in patients with a IV grade of ATH than in patients with a III grade of hypertrophy. Our study, despite identifying statistically significant differences in the level of vitamin D between control and patients (*p* < 0.0001), did not identify a clear correlation between the low levels of this vitamin and the ATH scores. In any case, the differences in the clinical features between children with a vitamin D deficit compared to the ones with a normal value might support the theory of the vitamin D role. The absence of a statistically significant *p*-value could be related to the small sample analyzed and the high variability of the vitamin D levels.

Vitamin D has a “multidirectional role”. The vitamin impacts the immune answer due to the activation of the enzyme 1-alfa-hydroxylase (CYP27B1) stimulating the innate and adaptive immunity against various types of pathogens [24] and modulating the gut and oral microbiota [25]. The combination of these effects improves the systemic and the local immune answer both in the gut and in the nose [26]. Recently, vitamin D has been identified as an immune-modulator and promoter of beneficial microbiota in the pharynx [27]. The nose as a barrier can interact with viruses, both fighting them and reducing their spread in the body. In the back of the nose (rhynopharinx), the adenoid tissue is located, which represents the first defensive barrier with the “lymphocyte soldier”, then descending in the oropharynx, we find the tonsils. The hypertrophy of these structures is meaningful of a correct answer to the viral infection.

Vitamin D regulates the expression of the active form of antimicrobial peptide cathelicidin, named LL-37, which is also an immune modulator. Elenius et al. found that higher serum LL-37 levels were associated with lower intratonsillar expression of Il-17, and serum LL-37 levels tended to increase with the increase of vitamin D levels [28]. The low level of vitamin D and the consequent inadequate modulation of the immune system might explain the compensative hypertrophy of the tonsils’ tissue observed in the upper aerodigestive tract. Sarmineto Varon et al. showed that tonsillar chronic inflammation and the alteration of oral microbiota could play a role in the pathogenesis of pediatric OSA [29]. Patients affected by OSA tend to suffer from daytime fatigue that might decrease the outdoor activity with the consequent reduced synthesis of vitamin D [30].

Several studies showed how increased BMI is associated with low levels of 25-OHD in children; in fact, this vitamin is available in the blood in patients with obesity because it remains tied to the fat cells [31,32]. According to the meta-analysis from Li et al. [33], the serum 25-OHD levels decreased with increasing OSASH severity; both overweight and obese OSA-patients showed low levels of vitamin D. Our study cannot confirm Li et al.’s data because we only had few overweight patients in both groups (n = 3, 6.5% in ATH group versus 0% in healthy patients), and no obese patients. The absence of obese patients in our study disagrees with previous studies in which a consistent number of overweight patients were suffering from OSA and/or hypovitaminosis D.

We observed more episodes of URIs per year in the ATH group than in the control group (212 vs. 89); considering the homogeneous distribution of the children in the two groups (66 vs. 64), we speculate that the lower levels of vitamin D might expose more to the recurrent infection of the upper respiratory tract.

Finally, we want to underline that the study was conducted in the winter, the period of minor exposure to the sun for people, and nevertheless we identified statistically significant differences between patients and control; we speculate that this might be related to an inherited alteration of some enzymes involved in 25-OHD synthesis (as CYP2R1) and a polymorphism in genes involved in vitamin D synthesis, hydroxylation, and transport [34].

We should also consider that vitamin D is implicated in the different neurochemical mechanisms involved in sleep regulation and mainly in the serotoninergic and dopaminergic pathways.

Vitamin D plays a central role in the regulation of serotonergic pathways, in melatonin production (which is essential for the sleep cycle regulation) by the regulation of the conversion of tryptophan into 5-HTP, and in the correct expression of VDRs in the prefrontal cortex and in limbic structures [35,36]; the presence of VDRs in limbic structures, including the hippocampus, amygdala, and prefrontal cortex, suggests that vitamin D could also be associated with the regulation of mood and emotional behavior. The vitamin could influence the serotoninergic pathway in the brain and in the peripheral tissues binding the vitamin D response elements (VDREs) on the tryptophan hydroxylase genes (THP1 and THP2), involved in serotonin production. Vitamin D inhibits the expression of THP1 in the peripheral tissues and increases the expression of THP2 in the brain.

On the other hand, VDRs are present in the human substantial nigra, and play an important function in the development and regulation of the dopaminergic system [37]; in light of these suggestions, adult patients return to normal sleep cycles with vitamin D levels at 60–80 ng/mL suggesting the need to reach levels higher than the normal accepted values of 30 ng/mL for the treatment of sleep disorders.

### Limits of the Study

The main limitation of the present work is the small size of the cohort that allows us to present only preliminary results; larger sample of patients are needed to confirm our data. Secondly, we did not collect data on the parathyroid hormone and other biomarkers of bone metabolism (calcium, 1,25-OHD serum level); however, we followed current paediatric guidelines that recommend measuring the 25-OHD serum levels to assess the vitamin D status [38].

On the other hand, this study presents several strengths that make our data very promising and worthy of additional and more extensive studies, as, first, this was a prospective study with a good matching between the control and disease group of age, sex and characteristics. Then, the exclusion of patients affected by allergies reduced the confounder. Finally, we conducted a strong statistical analysis to support our results.

## 5. Conclusions

We identified statistically significant differences between the ATH group and control, that, despite not being directly linkable to the lymphoid tissue hypertrophy (non-statistically significant *p*), could indirectly explain the effect of low vitamin D level on the immune answer. Because vitamin D acts at different levels, its blood concentration should be monitored, and if deficient, a supplement to increase its level should be prescribed. In fact, children with ATH and OSAHS suffer from daytime tiredness, which, by reducing the time of outdoor activity and sun exposure, could ulteriorly decrease the vitamin levels.

Vitamin D supplementation in childhood might be a good option to preserve the health and quality of life of these patients. Additional studies on larger sample are necessary to confirm our data.

## Figures and Tables

**Figure 1 children-10-00274-f001:**
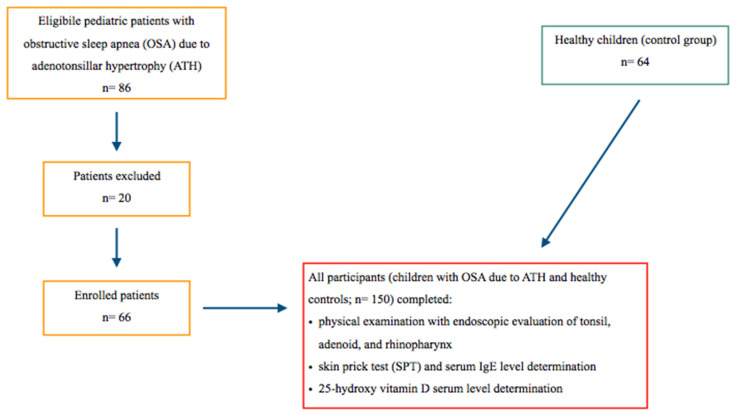
Study diagram flow-chart.

**Table 1 children-10-00274-t001:** Results of the home sleep apnea test.

**AHI**	Mean	9.41
	Median	6.7
	Range	3.9–42.1
**ODI**	Mean	7.39
	Median	6.2
	Range	2–43.9
**Nadir**	Mean	79%
	Median	80%
	Range	53–91%

## Data Availability

Not applicable.

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
