# Peer review of "Vitamin D Deficit as Inducer of Adenotonsillar Hypertrophy in Children with Obstructive Sleep Apnea—A Prospective Case-Control Study"

_children, 2023, doi:10.3390/children10020274_

Round 1

Reviewer 1 Report

- The title should be rewrite, because it's too hard to follow.

- Fix the writing, it's still messy (align right and left). As an example, the writing of the abstract that appears in the pdf file is still uneven and untidy.

- Include name and ethical approval number for this study.

- Please, provide study design diagrams to make it easier to understand this research.

Author Response

Dear Reviewer,

Thanks for reviewing our manuscript and for the valuable comments that helped us clarify some relevant aspects that were missed or unclear in the first version of the paper. We have read theyour comments and made the changes to address comments and concerns. We hope that the changes made in the revised manuscript and responses provided below have adequately addressed the reviewer’s comments and made this paper stronger.

1. The title should be rewrite, because it's too hard to follow.

Thanks for your comment. We rewrote the title, we sincerely hope that this new one can represent better the meaning of our paper.

2. Fix the writing, it's still messy (align right and left). As an example, the writing of the abstract that appears in the pdf file is still uneven and untidy.

Thanks; we fixed it.

3. Include name and ethical approval number for this study.

Thanks; we included the name of the Ethical Committee, the code, and the date of the approval.

4. Please, provide study design diagrams to make it easier to understand this research.

Thanks; you will find a diagram of the study flow-chart.

We really appreciated your careful and thoughtful evaluation of our manuscript and hope that this revised version meets with your approval. We have tracked all changes using the “track changes” tool of Microsoft Word. Thanks again for your interest in our work. We await your review of our revised manuscript.

Sincerely yours,

The Authors

Reviewer 2 Report

This is very interesting study but the main flaws are very low quality of study reporting and misleading terminology related to sleep medicine.

Major flaws:

1. Authors have to report this study strictly in accordance to the STROBE Statement (please use STROBE checklist) https://www.equator-network.org/reporting-guidelines/strobe/ Authors have to write that they used STROBE statement for study reporting.

2. Terminology: Authors used term "obstructive sleep apnea/hypopnea syndrome (OSAHS)". Are you sure that you diagnosed OSAHS not OSA? We use OSAHS only when OSA is associated with symptoms during the daytime (e.g. excessive daytime sleepiness, decreased cognitive function). Did you collect data related to daytime? This issue has to be precisely explain and correct term has to be use within title, abstract, keywords and manuscript body.

3. Materials and methods: Authors used term "domiciliary polysomnography" which is not correct term to describe home sleep apnea test (HSAT). We can also use term "home respiratory polygraphy". Authors have to revise it. Furthermore Authors have to provide details of used device, describe procedure, software, and scoring (using international guidelines and cite that guidelines).

Please describe procedure of blood samples collecting.

Please add a neoplasm disease and severe mental disorders to exclusion criteria. I think that Authors did not include children with neoplasm and severe mental disorders.

Please provide ID for the approval of Institutional Review Board.

4. Introduction: Authors have to write that vitamin D level is related to disorders of oral cavity and/or throat. I suggest the following latest and reliable articles:

Zakeri M, Parsian H, Bijani A, Shirzad A, Neamati N. Serum levels of vitamin D in patients with recurrent aphthous stomatitis. Dent Med Probl. 2021;58(1):27–30. doi:10.17219/dmp/126360

Krawiec M, Dominiak M. Prospective evaluation of vitamin D levels in dental treated patients: A screening study. Dent Med Probl. 2021;58(3):321–326. doi:10.17219/dmp/134911

Authors have to write that many disorders related to oral cavity are linked with sleeping habits in children. I suggest the following latest paper:

Topaloglu-Ak A, Kurtulmus H, Basa S, Sabuncuoglu O. Can sleeping habits be associated with sleep bruxism, temporomandibular disorders and dental caries among children? Dent Med Probl. 2022;59(4):517–522. doi:10.17219/dmp/150615

Authors have to add a paragraph why their study is novel and important before aim of the study.

Minor flaws:

1. Abstract: please explain each abbreviation before first use eg. OSAHS, ATH, CD. Please write OSAHS not OSAHA. The abstract is not done correctly in the graphical context, please correct it.

2. Authors have to revise the language of manuscript with native speaker.

Author Response

Dear Reviewer,

Thanks for reviewing our manuscript and for the valuable comments that helped us clarify some relevant aspects that were missed or unclear in the first version of the paper. We have read your comments and made the changes to address comments and concerns. We hope that the changes made in the revised manuscript and responses provided below have adequately addressed the reviewer’s comments and made this paper stronger.

Major flaws:

1. Authors have to report this study strictly in accordance to the STROBE Statement (please use STROBE checklist) https://www.equator-network.org/reporting-guidelines/strobe/ Authors have to write that they used STROBE statement for study reporting.

Thanks for your comment; we agree, and we add the sentence.

2. Terminology: Authors used term "obstructive sleep apnea/hypopnea syndrome (OSAHS)". Are you sure that you diagnosed OSAHS not OSA? We use OSAHS only when OSA is associated with symptoms during the daytime (e.g. excessive daytime sleepiness, decreased cognitive function). Did you collect data related to daytime? This issue has to be precisely explain and correct term has to be use within title, abstract, keywords and manuscript body.

Thanks for your valuable comment; as you suggest, we corrected it in the manuscript.

3. Materials and methods: Authors used term "domiciliary polysomnography" which is not correct term to describe home sleep apnea test (HSAT). We can also use term "home respiratory polygraphy". Authors have to revise it. Furthermore Authors have to provide details of used device, describe procedure, software, and scoring (using international guidelines and cite that guidelines).

Thanks; we provided the informations requested.

Please describe procedure of blood samples collecting.

Thanks; we explained how the blood sample were collected and processated.

Please add a neoplasm disease and severe mental disorders to exclusion criteria. I think that Authors did not include children with neoplasm and severe mental disorders.

Thanks; we agree, and we fixed it.

Please provide ID for the approval of Institutional Review Board.

Thanks; we add name, id, and date.

4. Introduction: Authors have to write that vitamin D level is related to disorders of oral cavity and/or throat. I suggest the following latest and reliable articles:

Zakeri M, Parsian H, Bijani A, Shirzad A, Neamati N. Serum levels of vitamin D in patients with recurrent aphthous stomatitis. Dent Med Probl. 2021;58(1):27–30. doi:10.17219/dmp/126360

Krawiec M, Dominiak M. Prospective evaluation of vitamin D levels in dental treated patients: A screening study. Dent Med Probl. 2021;58(3):321–326. doi:10.17219/dmp/134911

Thanks, we agree; as you suggested, we wrote this in the introduction with the reference you suggested. 

Authors have to write that many disorders related to oral cavity are linked with sleeping habits in children. I suggest the following latest paper:

Topaloglu-Ak A, Kurtulmus H, Basa S, Sabuncuoglu O. Can sleeping habits be associated with sleep bruxism, temporomandibular disorders and dental caries among children? Dent Med Probl. 2022;59(4):517–522. doi:10.17219/dmp/150615

Thanks, we agree; as you suggested, we wrote this in the introduction with the reference you suggested. 

Authors have to add a paragraph why their study is novel and important before aim of the study.

Thanks, we agree; we added this in the introduction before the aim of the study.

Minor flaws:

1. Abstract: please explain each abbreviation before first use eg. OSAHS, ATH, CD. Please write OSAHS not OSAHA. The abstract is not done correctly in the graphical context, please correct it.

Thanks, we fixed it.

2. Authors have to revise the language of the manuscript with native speaker.

Thanks, we agree; we revised the English language.

We really appreciated your careful and thoughtful evaluation of our manuscript and hope that this revised version meets with your approval. We have tracked all changes using the “track changes” tool of Microsoft Word. Thanks again for your interest in our work. We await your review of our revised manuscript.

Sincerely yours,

The Authors

Round 2

Reviewer 2 Report

I don't have further comments.

Author Response

Thanks